# Nutrient restriction enhances the proliferative potential of cells lacking the tumor suppressor PTEN in mitotic tissues

Katarzyna Nowak[†], Gerhard Seisenbacher[†‡], Ernst Hafen, Hugo Stocker[*]

Institute of Molecular Systems Biology, ETH Zürich, Zürich, Switzerland

**Abstract** How single cells in a mitotic tissue progressively acquire hallmarks of cancer is poorly understood. We exploited mitotic recombination in developing *Drosophila* imaginal tissues to analyze the behavior of cells devoid of the tumor suppressor PTEN, a negative regulator of PI3K signaling, under varying nutritional conditions. Cells lacking PTEN strongly overproliferated specifically in nutrient restricted larvae. Although the *PTEN* mutant cells were sensitive to starvation, they successfully competed with neighboring cells by autonomous and non-autonomous mechanisms distinct from cell competition. The overgrowth was strictly dependent on the activity of the downstream components Akt/PKB and TORC1, and a reduction in amino acid uptake by reducing the levels of the amino acid transporter Slimfast caused clones of *PTEN* mutant cells to collapse. Our findings demonstrate how limiting nutritional conditions impact on cells lacking the tumor suppressor PTEN to cause hyperplastic overgrowth.

**\*For correspondence:** stocker@ imsb.biol.ethz.ch

[†]These authors contributed equally to this work

[‡]**Present address:** Department of Experimental and Health Sciences, Universitat Pompeu Fabra, Barcelona, Spain

**Competing interests:** The authors declare that no competing interests exist.

**Reviewing editor**: David M Sabatini, Whitehead Institute/ Massachusetts Institute of Technology, United States

## Introduction

Clinically detectable cancer cells carry a multitude of mutations and chromosomal aberrations, and they display an enormous genetic heterogeneity (*Salk et al., 2010*; *Wong et al., 2011*; *Brosnan and Iacobuzio-Donahue, 2012*; *Marusyk et al., 2012*; *Turner and Reis-Filho, 2012*). It is therefore desirable to target earlier tumorigenic stages but we know comparatively little about how pre-cancerous cells progressively develop into tumors (*Moreno, 2008*). The model system *Drosophila* allows analyzing the behavior of cells lacking particular tumor suppressor functions. During the growth phase (larval instars), the cells of the imaginal discs (that will eventually give rise to adult appendages) remain diploid and proliferate until the discs have reached an appropriate size. The simple architecture of the imaginal discs (the disc proper consists of a single cell-layered epithelium and is covered by the peripodial epithelium) enables the labeling and tracking of cell populations. These cell populations can be genetically manipulated with the help of sophisticated tools. Finally, since the larvae live in their food, cellular stress situations can be imposed by controlling the food source.

We have focused our analysis on cells lacking the tumor suppressor PTEN (phosphatase and tensin homolog deleted on chromosome 10). PTEN is well conserved from flies to humans, and it is the second most frequently mutated tumor suppressor found in many types of human cancers (*Goberdhan and Wilson, 2003*; *Salmena et al., 2008*; *Hollander et al., 2011*; *Song et al., 2012*). PTEN antagonizes the function of the lipid kinase Phosphatidylinositide 3-kinase (PI3K); therefore, in the absence of PTEN, high levels of the lipid second messenger PIP3 result in an increased membrane recruitment and activation of the serine/threonine kinase PKB (protein kinase B, also known as Akt), which leads to enhanced cellular growth, proliferation, and survival (*Altomare and Testa, 2005*; *Georgescu, 2010*; *Song et al., 2012*). The consequences of activating PI3K signaling due to *PI3K* overexpression or loss of *PTEN* function has been extensively studied in *Drosophila* (*Leevers et al., 1996*; *Goberdhan et al., 1999*; *Huang et al., 1999*; *Weinkove et al., 1999*; *Gao et al., 2000*; *Britton et al., 2002*). Cells overexpressing *Dp110/PI3K* are enlarged and, in the fat body, increase their nutrient storage.

**eLife digest** Mutations are permanent changes to a cell's genome. If one or more mutations result in a cell proliferating in an unregulated manner, it is referred to as a cancer cell. The generation of cancer cells is a relatively common occurrence within organisms, but these rogue cells are generally recognized and destroyed by the organism's immune system. However, when the immune system fails to identify and eliminate cancer cells, they can proliferate to form malignant, life-threatening tumors.

Mutations in a gene called *PTEN* are often found within cells that develop into cancerous tumors. This gene is normally expressed as a protein that is involved in the regulation of cell division, preventing cells from growing and dividing too quickly. However, when the protein PTEN is absent or non-functional, cells experience enhanced growth, proliferation, and survival. Such cells are also thought to be resistant to nutrient restriction, but the mechanism responsible for this resistance is not well understood.

Here, Nowak et al. investigate the behavior of cells lacking PTEN in a fly model under a variety of nutritional conditions. When the supply of nutrients is limited, cells lacking PTEN shift resources from cell growth to cell multiplication. This appears to allow PTEN-deficient cells to outcompete neighboring wild-type cells; Nowak et al. suggest these rapidly proliferating cells are capable of effectively hoarding nutrient stores, both in their immediate vicinity and organism-wide. Further studies that focus on changes in gene expression may be able to uncover the mechanism that allows PTEN-deficient cells to proliferate when nutrients are restricted. Moreover, by shedding light on a factor that has an important influence on tumor development, these results may have implications for cancer treatment strategies.

This stockpiling of nutrients helps them to cell-autonomously bypass the nutritional requirements for cellular growth and DNA replication during amino acid deprivation (*Britton et al., 2002*). In mitotic tissues, clones of *PTEN* mutant cells are enlarged, which is mainly caused by an increase in cell size (*Leevers et al., 1996*). However, given the importance of PTEN as a tumor suppressor, the overgrowth caused by the loss of PTEN is rather mild (*Goberdhan et al., 1999*; *Huang et al., 1999*; *Gao et al., 2000*).

Recently, it has been demonstrated that tumors lacking PTEN or with increased PI3K activity are resistant to dietary restriction (*Kalaany and Sabatini, 2009*). This observation underscores the importance of understanding the intrinsic changes in early tumors caused by the microenvironment. Furthermore, it remains largely unknown how a growing tumor impacts on its environment.

In this study, we attempted to mimic early events in tumor development by inducing clones of *PTEN* mutant cells under conditions in which nutrients become limiting. We show that cells lacking PTEN switch from hypertrophic growth to hyperplastic growth under nutrient restriction (NR). This hyperproliferation occurs at the expense of neighboring wild-type cells, probably by competition for local and systemic pools of nutrients and other growth-promoting factors.

## Results

### Reduced insulin signaling is required for a proper starvation response

To assess the impact of starvation on survival, developmental timing, and weight, we reared larvae on food with varying yeast content. Yeast is the main source for micronutrients and amino acids in standard fly media. We observed a three-phase starvation response (*Figure 1A,B*). Our standard medium contains 100 g/l yeast. Down to 50 g/l yeast, larvae can be regarded as fully fed since the developmental time, weight, and survival were not affected. At a yeast concentration range between 50 g/l and 10 g/l, the development was gradually delayed and the weight decreased. This phase can be regarded as mild starvation, since survival was not compromised down to 25 g/l yeast concentration. Reduction of the yeast concentration to and below 10 g/l increased the developmental time from first instar to pupa and severely decreased body weight. Since mortality was also increased, this phase is regarded as severe starvation. Based on these results, the turning point between mild starvation and severe starvation is approximately at 10 g/l concentration of yeast.

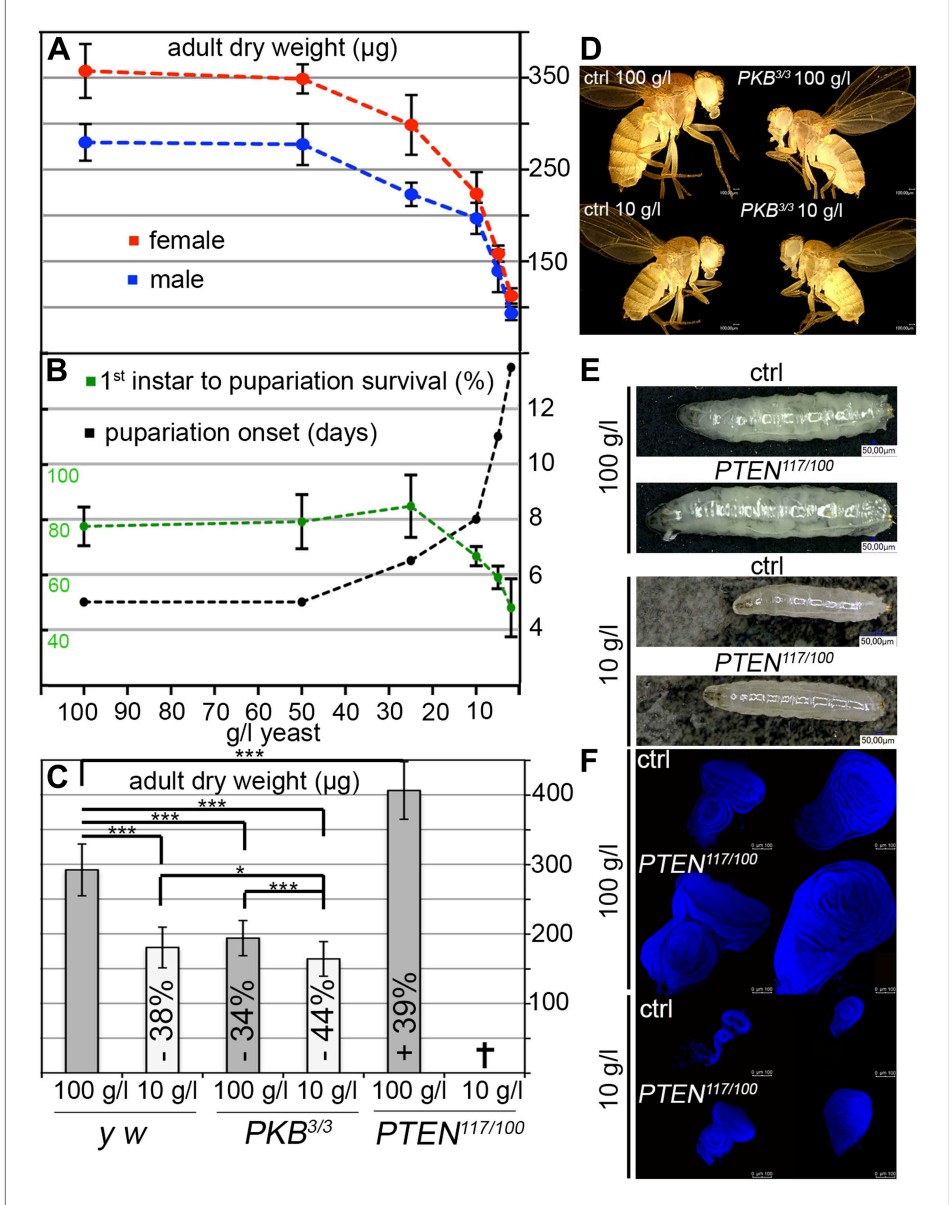

**Figure 1**. Response to yeast starvation in wild-type and insulin signaling defective *Drosophila*. (**A** and **B**) Control flies reared on standard culture medium with varying yeast concentrations. (**A**) Weight reduction in males (blue line) and females (red line). (**B**) Time from first instar to pupariation (black line) and the survival rates of first instar larvae to pupariation (green line). (**C**) Adult dry weight of wild-type, *PKB* and *PTEN* mutants reared on 100 g/l and 10 g/l yeast food, respectively. Weight difference (in %) with respect to control (*y w*) on 100 g/l yeast food is depicted in each column. (**D**) Size comparison of female control flies (*y w*) and *PKB* hypomorphic mutants reared on 100 g/l and 10 g/l food, respectively. (**E**) Larvae of *y w* and a *PTEN* hypomorphic combination (*PTEN¹¹⁷/PTEN¹⁰⁰*) reared on 100 g/l (late L3) and 10 g/l (mid L3). (**F**) Eye and wing discs of the larvae from (**E**).

The following source data are available for figure 1:

**Source data 1**. Adult dry weight; survival L1 to pupariation; pupariation time; weight analysis of IIS mutants.

We next subjected hypomorphic *PKB* and *PTEN* mutants to 100 g/l and 10 g/l yeast food, respectively (**Figure 1C–F**). Interestingly, fully fed hypomorphic *PKB* mutants weighed approximately the same as wild-type flies reared on 10 g/l yeast food. When reared on 10 g/l yeast food, *PKB* mutants were further delayed but only mildly decreased in size and weight (**Figure 1C,D**). In contrast, hypomorphic

*PTEN* larvae and adult flies were larger than wild-type controls under normal food conditions (*Figure 1E*). The prospective adult tissues, the eye and wing imaginal discs, were larger compared to control discs under both conditions (*Figure 1F*). *PTEN* hypomorphic larvae were highly sensitive to a reduction in yeast content, and although still larger, they did not survive to pupariation when reared on 10 g/l yeast food (*Figure 1C*). This is consistent with the findings that larvae with randomly induced PI3K overexpression clones are starvation sensitive (*Britton et al., 2002*). Thus, under NR conditions, insulin signaling needs to be dampened to allow for survival of the organism.

## Imaginal cells lacking PTEN gain a strong proliferative advantage under starvation conditions

In contrast to the situation at the organismal level, tumors with elevated PI3K activity are starvation resistant (*Kalaany and Sabatini, 2009*). To mimic the clonal nature of cancer, we analyzed the loss of *PTEN* function in clones generated in mitotic tissues (imaginal discs) by hsFlp/FRT-mediated mitotic recombination (*Golic and Lindquist, 1989*; *Xu and Rubin, 1993*). This system enables the generation of single genetically marked epithelial cells mutant for a defined tumor suppressor. Subsequent mitoses give rise to a patch of cells ('clone') devoid of the tumor suppressor. The clones can be tracked throughout development due to genetically encoded markers (usually GFP). In a typical experiment, we induced clones by a heat shock 24 hr after egg laying. Subsequently, the larvae were split into two populations and transferred to vials with yeast concentrations of 100 g/l and 10 g/l, respectively. The discs were then dissected to analyze the clones either at the same time point or at the same developmental stage (*Figure 2A*).

We monitored the growth behavior of clones of *PTEN* mutant cells (henceforward called *PTEN* clones) in larvae reared under decreasing yeast concentration in the culture medium. As previously published, *PTEN* clones were enlarged in fed larvae but they did not severely impact on the structure of the imaginal discs and of the adult eyes (*Goberdhan et al., 1999*; *Gao et al., 2000*) (*Figure 2B* and *Figure 2—figure supplement 1*). Under mild starvation conditions (30 g/l yeast), *PTEN* clones were further increased in size, resulting in an overall increase in disc size as compared to discs harboring control clones. Reducing the yeast concentration to 20 g/l and below led to a massive size increase of the *PTEN* clones and to a concomitant reduction of the surrounding tissue. Furthermore, the *PTEN* clones had fused and rendered the discs lobed in appearance (*Figure 2B* and *Figure 2—figure supplement 2*). At 5 g/l yeast concentration, the eye discs with *PTEN* mutant tissue were severely overgrown with polyp-like structures protruding from the discs (*Figure 2—figure supplement 2*). The resulting adult eyes displayed strong malformations and outgrowths, indicating a loss of epithelial integrity (*Figure 2—figure supplement 1*, red squares). In some cases, no clones could be recovered in the adult eyes, probably due to a collapse of the *PTEN* clones resulting in melanized scars (*Figure 2—figure supplement 3*, white arrow). Tangential eye sections did not reveal any structural defects in ommatidial arrangement (*Figure 2—figure supplement 4*). *PTEN* mutant cells overgrow in a cell-autonomous manner as the enlarged ommatidia were exclusively composed of mutant cells. The overgrowth phenotype of *PTEN* clones is not specific to the eye; it was also observed in other imaginal tissues like the wing disc (*Figure 2—figure supplement 5*).

Since starvation causes a developmental delay, the overgrowth of *PTEN* clones could be a consequence of the prolonged growth period. We therefore analyzed the *PTEN* clones at earlier time points (36 hr, 48 hr, and 60 hr after induction; *Figure 2C*). Already 36 hr after clone induction, the *PTEN* clones were increased in size, and the overgrowth of *PTEN* clones and the reduction in wild-type tissue were fully apparent after 60 hr (*Figure 2C*), excluding the developmental delay as a main cause of the overgrowth.

To investigate whether the early stages of starvation are crucial for the *PTEN* clones to overgrow, we performed 'food switch' experiments by transferring larvae with *PTEN* clones from starvation (10 g/l) to normal (100 g/l) food and vice versa at given time points. We used the eyeless-(ey)Flp/FRT system to induce clones specifically in the eye imaginal tissues early during larval development. When the transfer from starvation to normal food occurred within the first 72 hr, the overgrowth of the *PTEN* clones was efficiently rescued. Later shifts were no longer effective in suppressing the overgrowth. Consistently, it was sufficient to transfer larvae within the first 48 hr from normal to starvation food to produce the overgrowth phenotype. After this time point, *PTEN* clones did not acquire the full growth advantage over the surrounding tissue, confirming the importance of the initial stages for *PTEN* clones to develop the overgrowth phenotype (*Figure 2—figure supplement 6*).

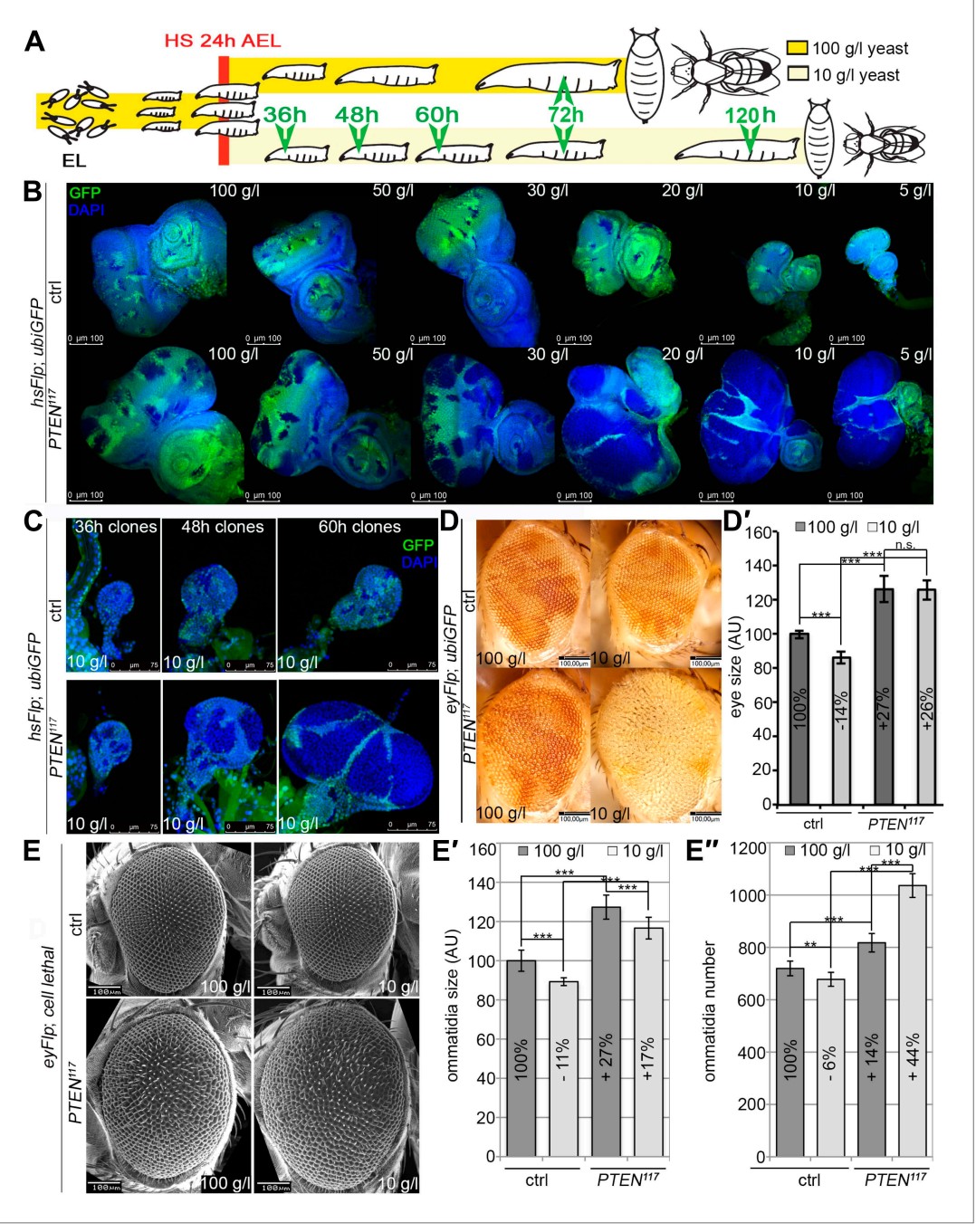

**Figure 2**. *PTEN* mutant cells are resistant to starvation and have a growth advantage upon NR. (**A**) Schematic drawing of the clonal induction with hsFlp/FRT-mediated mitotic recombination. After egg laying, larvae were allowed to grow for 24 hr on yeast paste. After the heat shock, larvae were distributed on different food conditions (e.g., 100 g/l and 10 g/l yeast). The time points of dissection are depicted in green. (**B**) Third instar eye discs bearing hsFlp/FRT *PTEN* mutant and control clones (marked by the absence of GFP) of larvae reared under varying yeast concentrations. (**C**) Eye discs with hsFlp/FRT *PTEN* mutant and control clones (marked by the absence of GFP) induced at the same time and conditions, reared on 10 g/l yeast food, and dissected at the indicated age of the clones. (**D**) Eyes bearing eyFlp/FRT *PTEN* mutant and control clones (marked by the absence of pigmentation) of animals reared on 100 g/l and 10 g/l yeast food, respectively. (**D′**) Quantification of the respective eye sizes. (**E**) Scanning electron micrographs of eyes almost exclusively composed of *PTEN* mutant or control tissue of animals reared on 100 g/l and 10 g/l food, respectively, and the quantification of ommatidia size (**E′**) and number (**E″**) from *PTEN* mutant and control eyes.

*Figure 2. Continued on next page*

*Figure 2. Continued*

The following source data and figure supplements are available for figure 2:

**Source data 1**. Eye size; eye measurements of SEM pictures.

**Figure supplement 1**. Severe starvation induces malformations in adult eyes bearing *PTEN* clones.

**Figure supplement 2**. Severe starvation affects architecture of discs bearing *PTEN* clones.

**Figure supplement 3**. Severely overgrown *PTEN* clones tend to collapse.

**Figure supplement 4**. *PTEN* mutant overgrowth is cell autonomous and does not affect differentiation.

**Figure supplement 5**. *PTEN* clones have a growth advantage in wing discs under starvation.

**Figure supplement 6**. *PTEN* mutant cells rapidly respond to the yeast content in the food.

We quantified the *PTEN* induced overgrowth by measuring the eye size of eyFlp/FRT-induced *PTEN* clones (*Figure 2D,D'*). Similar to the hsFlp/FRT-induced clones, the eyFlp/FRT-induced *PTEN* clones and the entire eyes were slightly enlarged in flies raised on normal food. Under starvation conditions, the mutant tissue occupied most of the adult eye and the wild-type tissue was severely reduced. This overrepresentation of the mutant tissue resulted in an absolute increase of eye size under starvation as compared to normal food conditions (*Figure 2D,D'*).

To investigate the effects of reduced nutrition on ommatidia size and number, we generated eyes almost completely composed of *PTEN* mutant tissue by means of the eyFlp/FRT cell lethal system and analyzed them by scanning electron microscopy (*Figure 2E–E''*). Under normal food conditions, the eye size increase was mainly caused by larger ommatidia (+27%), and to a minor extent by more ommatidia (+14%). Under 10 g/l yeast food, the ommatidial size was roughly proportionally reduced in wild-type and *PTEN* mutant eyes. Intriguingly, whereas the ommatidia number was decreased by 6% in wild-type eyes, it was massively increased in *PTEN* mutant eyes (+44%). Since the composition of the *PTEN* mutant ommatidia remained unchanged, the size and number of ommatidia reflects cell size and cell number. Thus, the *PTEN* mutant tissue displays a switch from hypertrophy to hyperplasia.

## The growth advantage of *PTEN* mutant cells is not due to cell competition

Cell competition, a process that results in the elimination of suboptimal cells from a growing tissue (*Morata and Ripoll, 1975*; *Simpson, 1979*; *Simpson and Morata, 1981*), could contribute to the overgrowth of *PTEN* clones. 'Winner' cells actively eliminate 'loser' cells by inducing apoptosis and thereby take over the tissue (*Moreno et al., 2002*; *Moreno and Basler, 2004*; *de la Cova et al., 2004*). If *PTEN* mutant cells acted as supercompetitors, the surrounding tissue would suffer from apoptosis. We therefore monitored apoptosis in eye imaginal discs bearing *PTEN* clones by cleaved Caspase-3 and TUNEL staining (*Figure 3A,B*). Whereas only few apoptotic cells were detected in the heterozygous tissue under both fed and starving conditions, increased levels of apoptosis were observed within the overgrowing *PTEN* clones (*Figure 3A,A'*).

Since the overgrowth is fully apparent at 60 hr after clone induction, we investigated *PTEN* clones at 48 hr to exclude the possibility that cell competition eliminates the surrounding tissue at this early stage (*Figure 3B*). Again, most of the apoptosis was observed in the mutant tissue, arguing against a major role of cell competition in the overgrowth phenotype. Expression of *p35* or of dominant negative JNK (*bsk^DN*; *Adachi-Yamada et al., 1999*) exclusively in the *PTEN* clones (by means of the MARCM system) efficiently blocked apoptosis and enhanced the overgrowth (*Figure 3C*, *Figure 3— figure supplement 1*, and not shown).

To further address the role of apoptosis in the surrounding tissue, we expressed *p35* in the dorsal half of the eye and randomly induced *PTEN* clones throughout the disc (*Figure 3D*). Consistent with the above results and the apoptosis pattern, *PTEN* clones overgrew even more in the dorsal compartment, and the surrounding tissue still got strongly reduced. Thus, the cells neighboring *PTEN* clones are not eliminated by apoptosis.

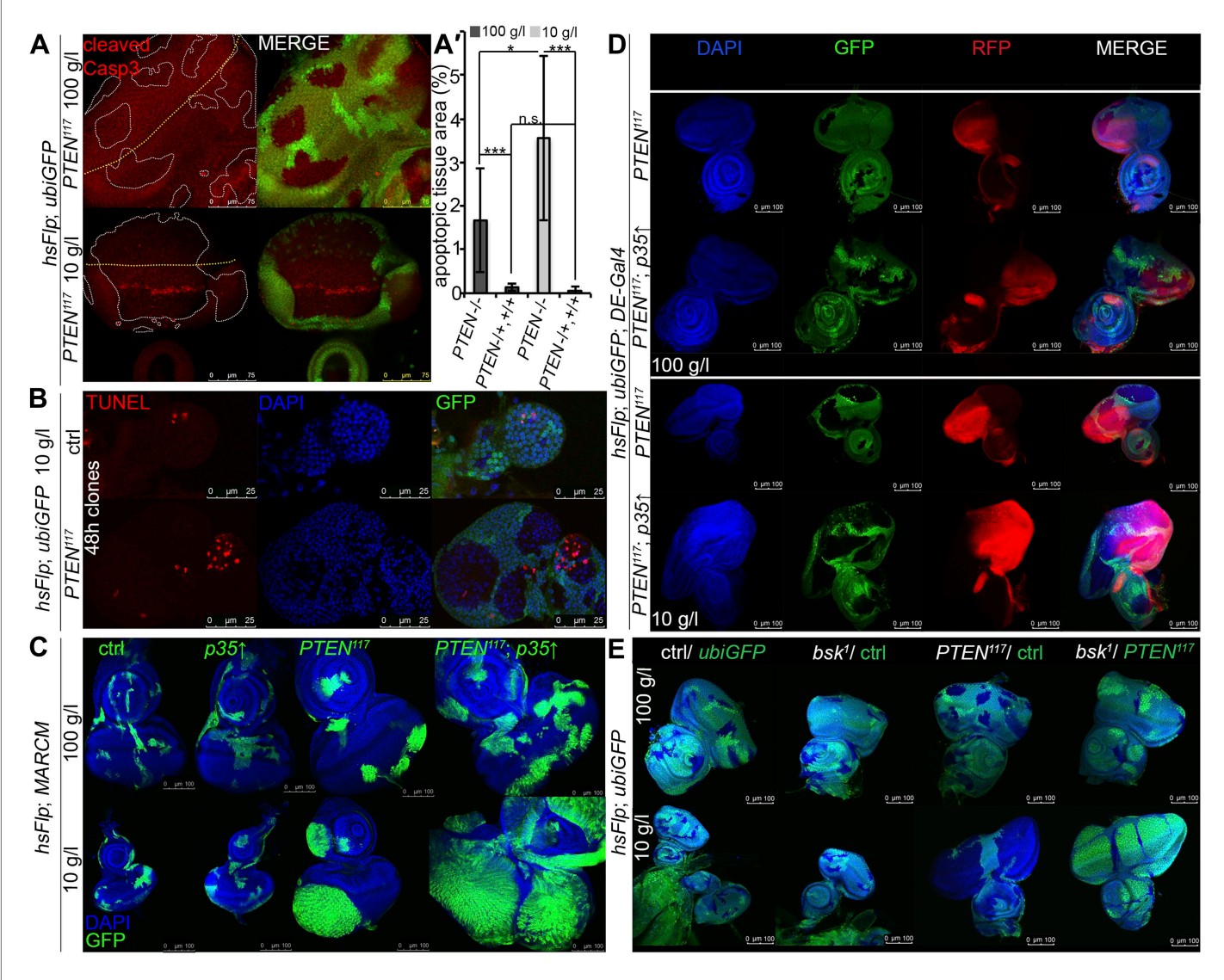

**Figure 3**. *PTEN* mutant cells are susceptible to cell death. (**A**) Cleaved Caspase-3 antibody staining (in red) of hsFlp/FRT *PTEN* clones (marked by the absence of GFP) in eye imaginal discs of larvae reared under the indicated conditions. The yellow dashed line indicates the morphogenetic furrow. About one third of the *PTEN* clones contain apoptotic cells. (**A'**) Quantification of the apoptotic tissue area (positive for cleaved Caspase-3 immunostaining) in the *PTEN* mutant and the surrounding tissue (*PTEN^{117}/+, +/+*), respectively, relative to the total area of the respective cell populations. (**B**) TUNEL staining (in red) on eye imaginal discs harboring 48 hr control and *PTEN* clones reveals high apoptotic levels within the *PTEN* clones but only residual apoptosis in surrounding tissue or control discs. (**C**) Expression of anti-apoptotic *p35* within *PTEN* clones (marked by GFP) enhances the overgrowth potential of *PTEN* mutant tissue. (**D**) Inhibition of apoptosis in the dorsal half of the eye discs by expression of *p35* under the control of *DE-Gal4* (marked by RFP, red) with randomly induced *PTEN* clones (marked by the absence of GFP) does not rescue the surrounding (GFP positive) tissue. (**E**) Inhibition of JNK-mediated apoptosis in the sister clones of the *PTEN* clones does not rescue the surrounding tissue. The different clones are marked as depicted with the colors in the labeling (green: GFP positive, white: GFP negative).

The following source data and figure supplements are available for figure 3:

**Source data 1**. Tissue size; apoptotic area; ratio apoptotic area/tissue size.

**Figure supplement 1**. Blocking cell death in *PTEN* clones enhances the overgrowth.

**Figure supplement 2**. The massive overgrowth of *PTEN* clones under harsh starvation conditions correlates with high levels of apoptosis.

We also tested for the requirement of JNK signaling in the tissue neighboring the *PTEN* clones. hsFlp/FRT clones of *bsk[1]* over a chromosome carrying a GFP marker and a *PTEN* mutation were induced. In this way, mitotic recombination events generate two adjacent twin spots mutant for *bsk[1]* and *PTEN,* respectively. If *PTEN* cells attained the growth advantage by inducing JNK-mediated cell death in the twin spot, the overgrowth would be suppressed and the twin spot should grow larger. However, *PTEN* clones were overgrown irrespective of blocking JNK signaling in neighboring cells (*Figure 3E*).

Under severe starvation (5 g/l yeast content), the *PTEN* mutant polyp-like structures outgrowing from the eye discs exhibited very high levels of apoptosis (*Figure 3—figure supplement 2*). In the corresponding adult eyes, we often observed scars that were probably left behind by the collapsing clones (*Figure 2—figure supplement 3*). Those eyes contained more wild-type cells, indicating that the lost *PTEN* mutant tissue was compensated for. Thus, *PTEN* mutant cells are running at the edge of survival under limited nutrient conditions. They start to disobey the organ boundaries but are prone to apoptosis.

## *PTEN* mutant cells have elevated PIP3 levels, highly active PKB, and require TORC1 activity to overgrow

To gain a better understanding of the downstream events contributing to the overgrowth of *PTEN* clones, we monitored the levels of the second messenger PIP3 in discs containing *PTEN* clones by means of the tGPH reporter, a GFP–PH domain fusion protein with high affinity for PIP3 (*Britton et al., 2002*) (*Figure 4A*). The GFP–PH domain fusion protein was strongly localized at the membrane under both fed and NR conditions in *PTEN* clones. In contrast, the reporter signal was diffuse in the cytoplasm in the surrounding tissue under both conditions. Removal of *Insulin receptor* (*InR*) function from the *PTEN* mutant tissue only slightly reduced the overgrowth. Thus, in the absence of PTEN, the cellular PIP3 levels are sufficient to sustain the overgrowth regardless of the upstream signaling (*Figure 4—figure supplement 1*).

Similarly, elevated levels of phosphorylated PKB (P-PKB) were found at the membrane in *PTEN* clones at 100 g/l and 10 g/l yeast content (*Figure 4B*). These results were further confirmed by a Western blot analysis comparing discs exclusively composed of *PTEN* mutant tissue with discs bearing wild-type clones under normal food conditions (*Figure 4—figure supplement 2*). Due to the extremely small size of the wild-type discs under NR, we were not able to generate enough tissue for the analysis. Although the total levels of PKB were decreased in *PTEN* mutant tissue under both conditions as compared to the control under normal food conditions, the P-PKB levels were elevated. Thus, high PIP3 levels result in robust PKB activation under standard and starvation conditions. Consistently, reducing or removing *PKB* function by using a PH-domain mutant (*Stocker et al., 2002*) and a kinase-dead PKB (*Staveley et al., 1998*), respectively, completely suppressed the *PTEN* overgrowth under normal as well as under starvation conditions (*Figure 4C*).

The transcription factor FoxO is an important target of PKB. Upon high PI3K signaling activity, FoxO is phosphorylated by PKB and sequestered in the cytoplasm, and FoxO-mediated transcription of growth-suppressing genes is thus inhibited (*Brunet et al., 1999*; *Kops et al., 1999*; *Burgering and Kops, 2002*). FoxO was localized in the cytoplasm in *PTEN* mutant tissue under both conditions, whereas starvation induced the shuttling of FoxO into nuclei in the control tissue (*Figure 4E*). It has been shown that overexpression of *FoxO* decreases eye size, which is exacerbated under starvation conditions (*Junger et al., 2003*; *Kramer et al., 2003*). Consistent with the inactivation of FoxO in *PTEN* mutant cells, the FoxO-induced eye size reduction was not observed in *PTEN* clones (*Figure 4D* and *Figure 4—figure supplements 3 and 4*).

Signaling via Target of Rapamycin complex 1 (TORC1) promotes growth in response to nutrients. High TORC1 activity boosts cellular growth, at least in part via the phosphorylation of S6K and 4E-BP (reviewed in *Hietakangas and Cohen, 2009*). *PTEN* mutant cells did respond to starvation by reducing the levels of S6K and of phosphorylated S6K (P-S6K), as revealed by Western blot analysis on discs bearing eyFlp/FRT cell lethal clones of *PTEN* (*Figure 4—figure supplement 2*). PKB signaling activity was also slightly reduced as compared to standard conditions (*Figure 4B* and *Figure 4—figure supplement 2*). We next inhibited TORC1 activity selectively in *PTEN* clones by co-overexpression of Tuberous Sclerosis Complex-1 and -2 (Tsc1/2)—which together form a complex with GTPase activating protein (GAP) activity towards the small GTPase Rheb, an essential activator of TORC1 (*Garami et al., 2003*; *Zhang et al., 2003*)—or by removing *Rheb* function. Reducing TORC1 activity suppressed the

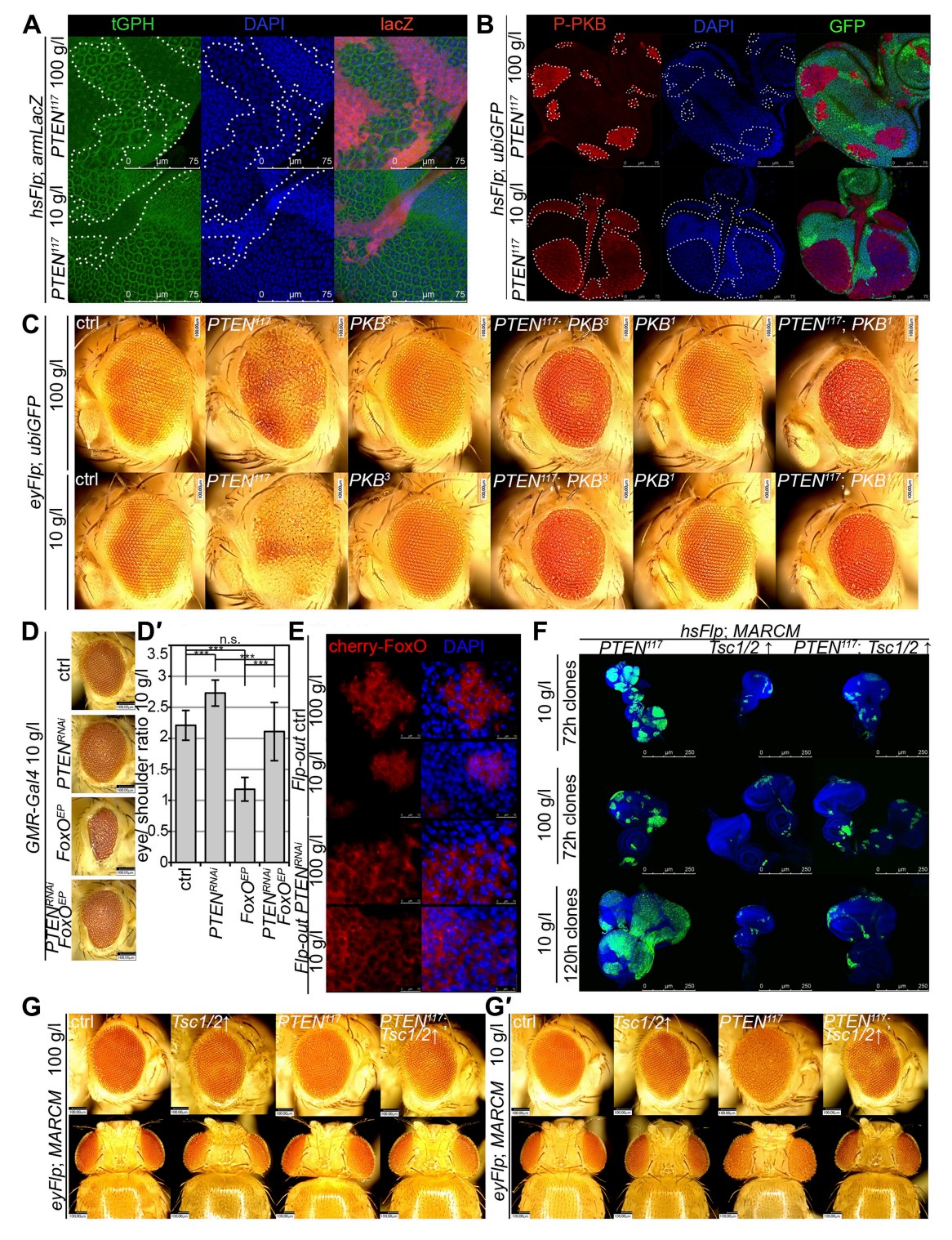

**Figure 4**. Interaction of insulin and TOR signaling with *PTEN* clones under starvation. (**A**) tGPH reporter (green) reveals high levels of PIP3 in *PTEN* clones (clones marked by absence of LacZ in red) under food conditions indicated. (**B**) P-PKB (red) is strongly increased in *PTEN* clones (marked by the absence of GFP) under normal conditions as well as under starvation. (**C**) *PTEN* overgrowth under normal and starvation conditions is strictly dependent on PKB activity. (**D**) Overexpression of *FoxO* results in a small eye under starvation conditions. This phenotype is suppressed by co-knockdown of *PTEN*. (**D'**) Quantification of eyes from (**D**). (**E**) Cherry-tagged FoxO localizes to the cytoplasm under normal food conditions and moves to the nucleus

*Figure 4. Continued on next page*

*Figure 4. Continued*

under starvation. This nuclear shuttling is prevented in *PTEN* clones. Clones are positively marked by Cherry (in red). (**F**) Inhibiting TORC1 activity by overexpression of *Tsc1/2* in *PTEN* clones (positively marked by GFP) suppresses the overgrowth. The suppression is evident already at 72 hr after clone induction. (**G**) Adult eyes showing the suppression of the overgrowth associated with *PTEN* clones by overexpression of *Tsc1/2* under starvation under standard conditions and under starvation (**G′**).

The following source data and figure supplements are available for figure 4:

**Source data 1**. Eye size; shoulder size; eye/shoulder ratio; ommatidia number.

**Figure supplement 1**. Genetic interactions between *PTEN* and IIS/TOR signaling components.

**Figure supplement 2**. Insulin and TOR signaling is modulated in *PTEN* clones in response to starvation.

**Figure supplement 3**. The effects of *PTEN* knockdown and *FoxO* overexpression neutralize each other.

**Figure supplement 4**. Overexpression of *FoxO* does not suppress *PTEN* mutant overgrowth.

**Figure supplement 5**. *PTEN* clones require TORC1 activity to overgrow.

**Figure supplement 6**. Reducing TORC1 activity suppresses *PTEN* mutant overgrowth.

**Figure supplement 7**. *PTEN* mutant cells require TORC1 activity to acquire a proliferative advantage under starvation.

overgrowth of the *PTEN* mutant tissue (***Figure 4F,G, 4G′*** and ***Figure 4—figure supplement 5***). P-PKB levels, however, were increased in the mutant tissue under both fed and NR conditions (***Figure 4—figure supplement 6***), which is in agreement with the negative feedback regulation of TORC1 on the activation of PKB (***Radimerski et al., 2002***; ***Kockel et al., 2010***). Eyes mutant for *PTEN* and a hypomorphic allele of *TOR* were reduced to the size of control eyes at both fed and starved conditions, indicating that the overgrowth of *PTEN* clones strictly depends on normal TORC1 function (***Figure 4—figure supplement 7***). Interestingly, whereas the size reduction of *PTEN* clones caused by impaired TORC1 function was primarily due to smaller ommatidia under normal conditions, ommatidia size was only slightly affected under starvation. By contrast, the hyperproliferation observed in *PTEN* clones under NR was almost completely abolished (***Figure 4—figure supplement 7***). Thus, TORC1 is indispensable for the switch to hyperproliferation observed in *PTEN* mutant tissue upon NR.

## Knockdown of the amino acid transporter Slimfast suppresses the overgrowth of *PTEN* mutant cells

The cationic amino acid transporter Slimfast (Slif) has been shown to activate TOR signaling in the fat body, and it appears to be an important component of a systemic nutrient sensor mechanism (***Colombani et al., 2003***). Slif function is also required in mitotic tissues, as clones in which *Slif* has been knocked down (*Slif^anti^*) are reduced in size, possibly due to reduced TORC1 activation. Since TORC1 is required for the overgrowth of *PTEN* clones, knocking down *Slif* should suppress this phenotype in a similar way as observed for *Tsc1/2* overexpression. Whereas expressing *Slif^anti^* in *PTEN* clones resulted in a slight reduction of the mutant tissue under normal conditions, it caused a nearly complete elimination of the *PTEN* mutant tissue under starvation conditions, while not affecting the control clones (***Figure 5A***). The loss of *PTEN* mutant tissue was accompanied by a massive increase in apoptosis, and GFP-positive (and thus *PTEN* mutant) cellular remnants were scattered throughout the disc (***Figure 5B***), suggesting that *PTEN* clones collapsed after an initial overgrowth. Indeed, early overgrowth of *PTEN* clones and high P-PKB signals were observed under starvation (***Figure 5C*** upper panel). Later, *Slif* function became limiting in *PTEN* clones that were subsequently eliminated (***Figure 5A,C*** lower panel). Intriguingly, the structure of the resulting adult eyes that had lost the *PTEN* clones was completely normal, indicating that the surrounding tissue was able to compensate for the loss of the clones (***Figure 5D,D′***). The effects that the reduction of Slimfast exerted on *PTEN* clones are distinct from those resulting from inhibiting TORC1 function, as overexpression of *Tsc1/2* suppressed the overgrowth of *PTEN* clones already at an early stage (***Figure 4F***). This may indicate that, rather

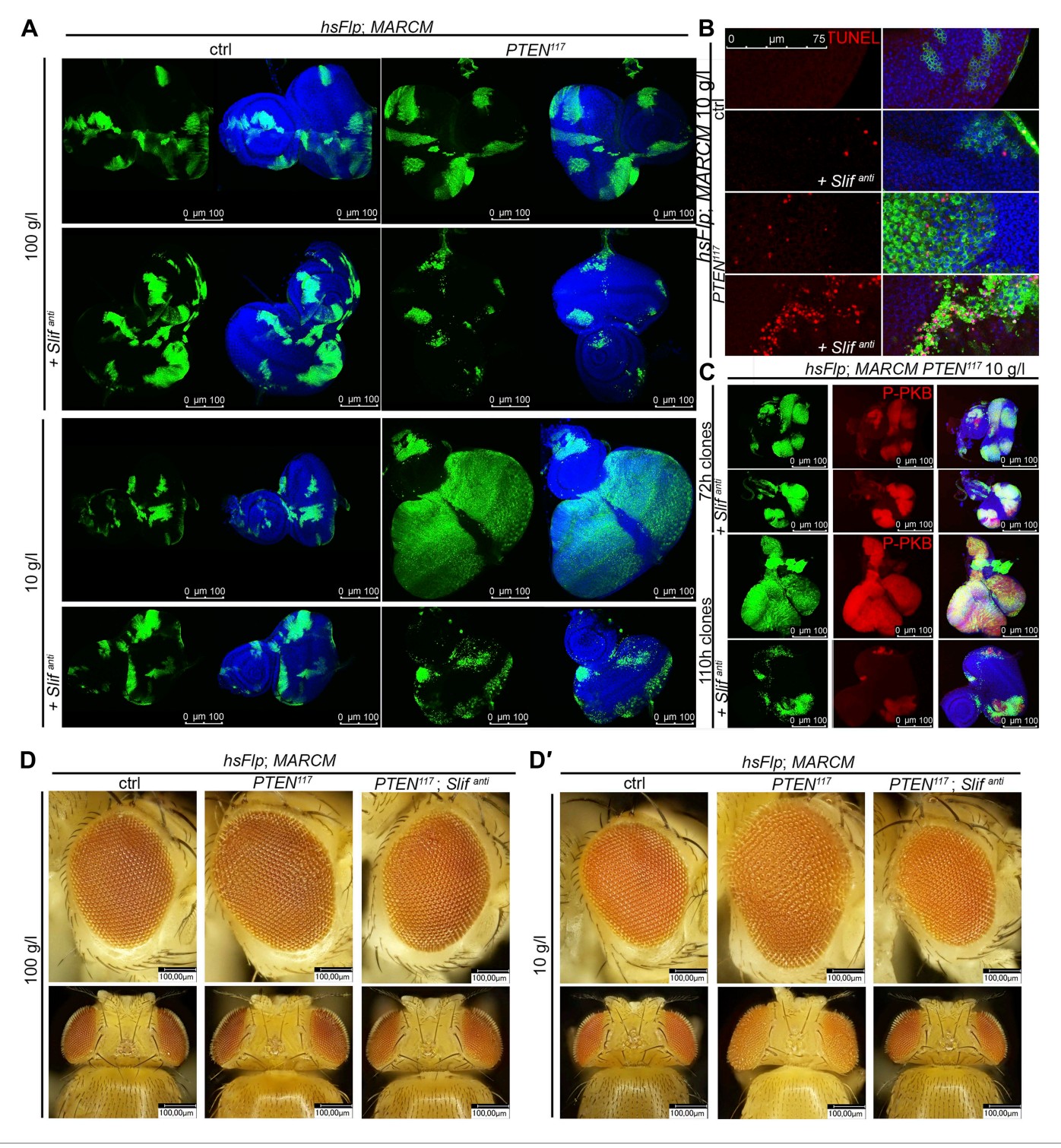

**Figure 5**. *PTEN* mutant tissue critically depends on the amino acid transporter Slimfast. (**A**) *PTEN* clones co-expressing *Slif^anti* collapse and disappear under starvation, whereas *Slif* reduction does not affect control clones. (**B**) TUNEL staining reveals high apoptosis levels (in red) in cells with *Slif^anti* expression. Reducing the levels of Slif eliminates *PTEN* mutant cells by apoptosis. (**C**) Dying 110 hr-old *PTEN* clones expressing *Slif^anti* lose the P-PKB signal (red), although they initially overproliferate (72 hr-old clones) indistinguishably from *PTEN* mutant tissue. (**D** and **D′**) Adult eyes with unmarked control, *PTEN* and *PTEN* plus *Slif^anti* clones under standard (**D**) and starvation conditions (**D′**). The reduction of Slif levels rescues the overgrowth associated with *PTEN* mutant eyes, especially under starvation conditions. All clones in the discs are positively marked by GFP.

*Figure 5. Continued on next page*

*Figure 5. Continued*

The following figure supplements are available for figure 5:

**Figure supplement 1**. Autophagy does not contribute to the initial survival of *PTEN* clones with reduced Slif levels under starvation.

**Figure supplement 2**. Inhibition of cell death in *PTEN* clones with reduced Slif prevents the initial collapsing in eye discs.

**Figure supplement 3**. Blocking autophagy and apoptosis in *PTEN* clones with reduced Slif does not prevent them from dying.

than the activation of TORC1, another aspect of Slif function—probably the amino acid influx itself—is critical for the survival of the *PTEN* mutant tissue.

We also assessed whether the initial overgrowth of the *PTEN* clones with reduced Slif function could be attributed to autophagy that maintains the levels of amino acids and thereby sustains TORC1 activity. However, reducing Atg5, a key component of the autophagic machinery, did not impact on the initial growth of *PTEN* clones with reduced Slif function (*Figure 5—figure supplements 1 and 3*). Blocking apoptosis (by means of *p35* expression) in *PTEN* clones with diminished Slif function delays the collapse of the clones as the clones can still be detected in imaginal discs (*Figure 5—figure supplements 1 and 2*) but are absent from adult eyes (*Figure 5—figure supplement 3*).

## *PTEN* clones affect the neighboring tissue and the organism in a non-autonomous way

We also observed non-autonomous effects of the *PTEN* clones on the surrounding tissue. P-PKB levels in the wild-type tissue neighboring with *PTEN* clones were reduced as compared to the tissue neighboring with control clones (*Figure 6—figure supplement 1*). This reduction is more apparent under normal conditions, which can be attributed to the already low P-PKB levels in discs of starved animals.

We were wondering whether this cell non-autonomous phenomenon only affects closely neighboring tissue or whether it acts at a longer range. Knocking down *PTEN* in the dorsal part of the eye not only resulted in its size increase, but also reduced the size of the ventral part as compared to control eyes (*Figure 6A,A'*). This growth-reducing effect within the eye was already visible under standard conditions but further enhanced on 10 g/l yeast food. In contrast to the null mutant situation, RNAi-mediated knockdown of *PTEN* did not result in enhanced overgrowth under starvation as compared to standard conditions, probably due to residual PTEN in the knockdown situation.

The analysis of flies with heads mostly composed of *PTEN* mutant tissue revealed similar non-autonomous effects on the sizes of other organs and of the entire body. A decrease in shoulder area (as a measure for body size) was observed under starvation but not under normal conditions (*Figure 6B*). Consistently, the wing size and the fat body nuclear size were reduced, especially in the animals that were raised under starvation (*Figure 6C,C',D,D'*). Thus, the growth-reducing non-autonomous effect of *PTEN* mutant tissue acts systemically.

The observed non-autonomous effects suggest that the *PTEN* mutant tissue efficiently competes with other larval tissues for common resources to support their massive growth, thereby launching a vicious cycle in the neighboring tissue that gets further starved and reduced. If our interpretation was correct, *PTEN* clones should overgrow less under starvation when growth signaling is maintained in neighboring cells. We attempted to promote IIS by expressing the ligand *Dilp-2* in eye imaginal discs (*Brogiolo et al., 2001*; *Ikeya et al., 2002*). Whereas expressing *Dilp-2* under *ey-Gal4* control did not affect control eyes, it suppressed the overgrowth caused by the loss of *PTEN* under starvation conditions. Furthermore, it restored the growth of both the tissue surrounding the *PTEN* clones and peripheral tissue (*Figure 7A*).

We also analyzed the consequences of introducing cells with deregulated TORC1 activity—and thus enhanced cellular growth–in the neighborhood of *PTEN* clones. To this end, we simultaneously generated *PTEN* clones and clones of *Tsc1* mutant cells (*Figure 7—figure supplement 1*). Interestingly, the mutant clones did not have additive effects on overgrowth. The introduction of *Tsc1* clones into the neighborhood of *PTEN* clones rather reduced their overgrowth under starvation. The discs were completely composed of either *Tsc1* or *PTEN* mutant tissue, as the clones did not heavily overlap. Thus, when neighboring with a tissue that also has a growth advantage, *PTEN* clones themselves experience competition for common resources.

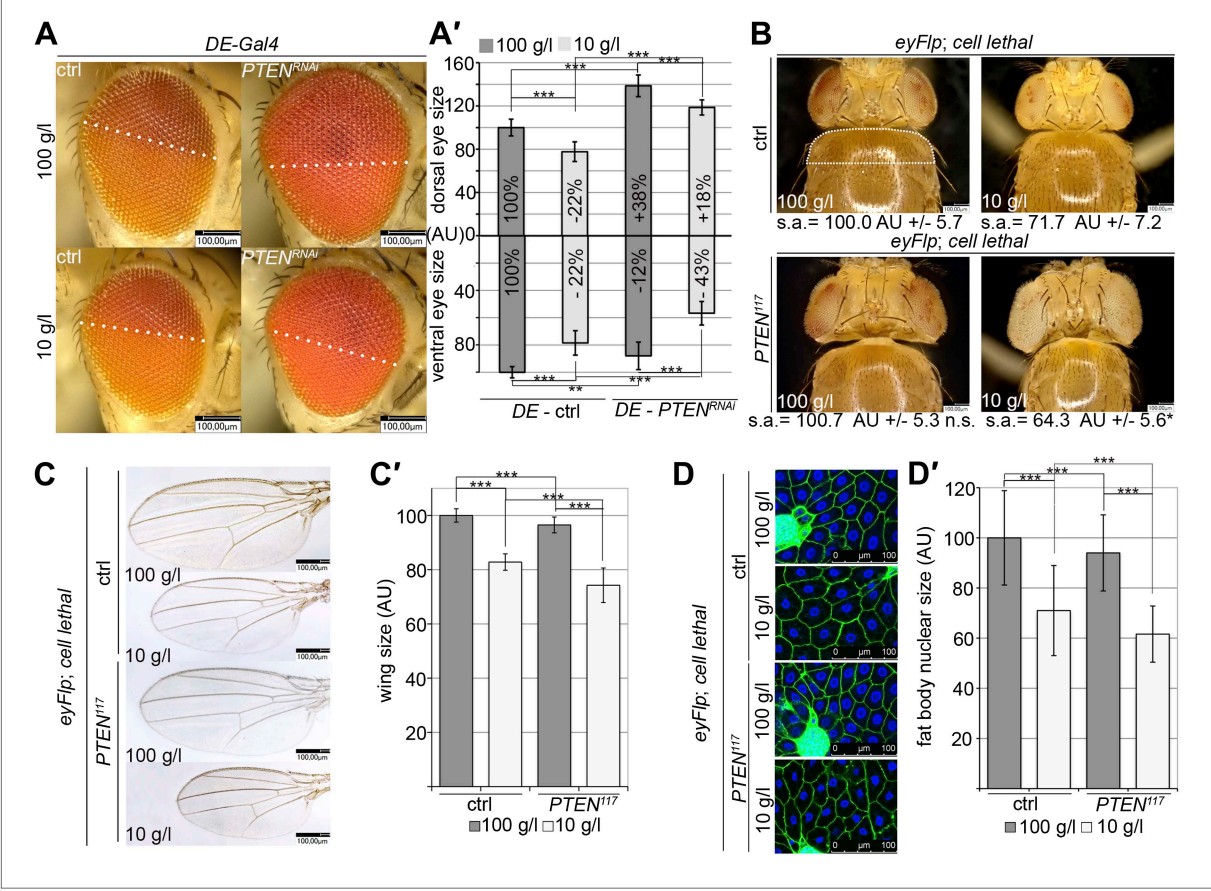

**Figure 6**. Systemic effects of *PTEN* clones on peripheral tissues. (**A**) *PTEN* was knocked down in the dorsal part of the eye by means of *DE-Gal4*. (**A'**) Quantification of the sizes of the ventral and dorsal halves. Percentage indicated in the bars represents the size reduction with respect to the respective part of control eyes under normal conditions. (**B**) Flies with *PTEN* mutant heads have smaller bodies as judged by shoulder area (s.a.) under starvation conditions. (**C**) Wings of flies with *PTEN* mutant heads under starvation are smaller than wings of control flies. (**C'**) Quantification of the wings from (**C**). (**D**) Size of fat body nuclei of larvae with *PTEN* mutant eye discs is decreased under starvation as compared to the control. (**D'**) Quantification of nuclear size from (**D**). *PTEN* mutant heads were generated by the eyFlp/FRT cell lethal system. Statistical analyses were done with Student's *t*-test (two tailed).

The following source data and figure supplements are available for figure 6:

**Source data 1**. Dorsal eye size; shoulder size; wing size; nuclear size in fat bodies.

**Figure supplement 1**. *PTEN* mutant tissue influences PKB signaling in the neighboring tissue.

To investigate whether the overgrowth of *PTEN* clones is enabled by a non-autonomous influence from the starved neighboring tissue, we genetically starved the surrounding tissue of *PTEN* clones by reducing TORC1 activity. We induced eyFlp/FRT clones of *PTEN* with sister clones mutant for *TOR* (*Figure 7B*, *Figure 7—figure supplements 2 and 3*). The resulting eyes were completely composed of *PTEN* mutant tissue already under standard conditions. However, the eyes were smaller as compared to eyes bearing *PTEN* clones only.

We also observed that clones of *TOR* mutant cells generated in otherwise wild-type eyes were massively underrepresented. The adult eyes were almost completely composed of wild-type twin clone tissue (*Figure 7B*), suggesting that not only *PTEN* mutant cells with a high growth potential but also wild-type cells neighboring with cells impaired in TORC1 function attain a growth advantage and proliferate in place of the slower growing cells. Similar observations were made with mutations impinging on IIS activity (e.g., *PKB*; *Figure 7—figure supplement 4*). Thus, relative differences in IIS activity cause a differential growth behavior.

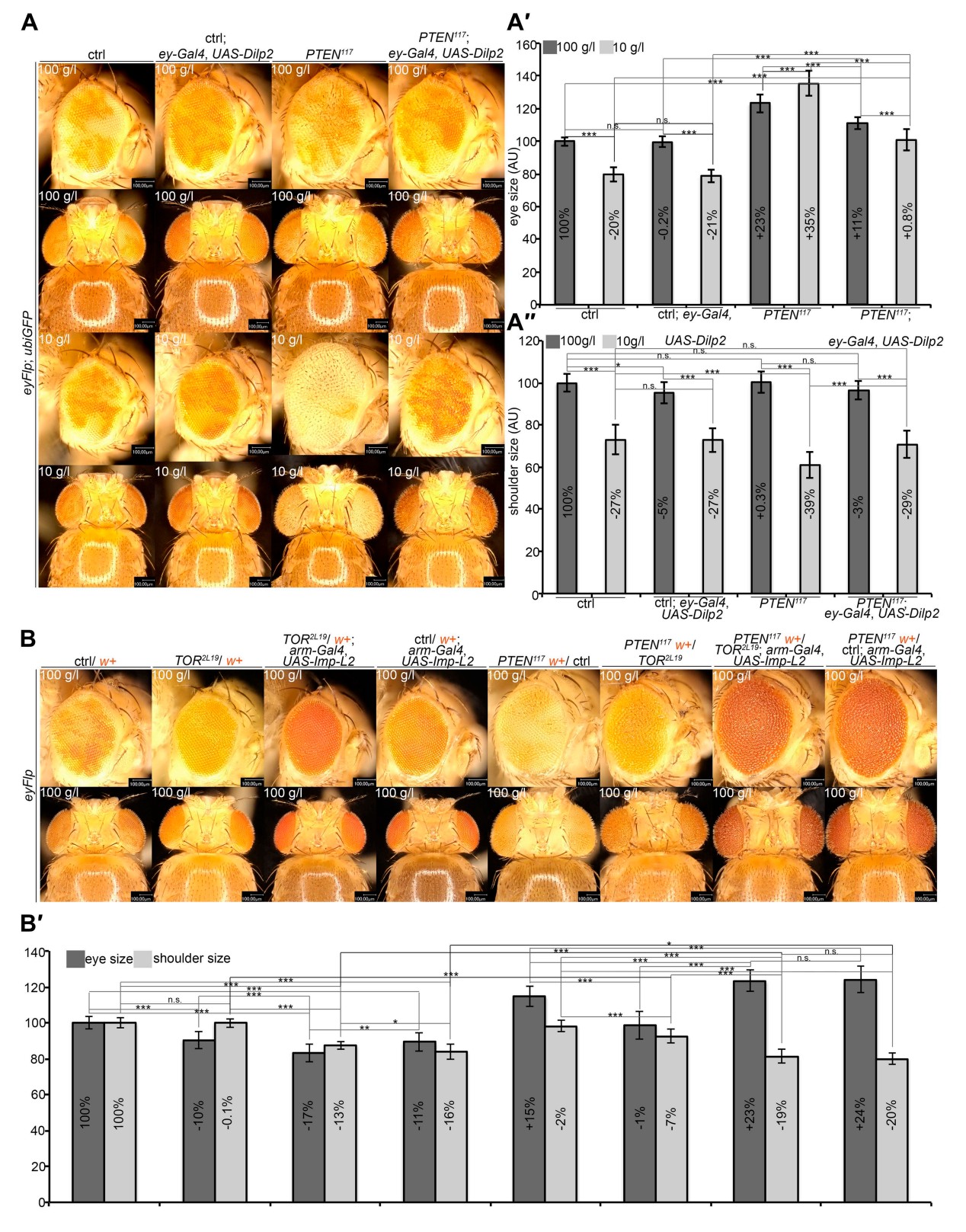

**Figure 7**. A reduction in growth signaling in the direct neighborhood and in peripheral tissues is required for the overgrowth of *PTEN* mutant cells. (**A**) Restoring insulin signaling by means of *Dilp-2* expression (driven by *ey-Gal4*) suppresses the overgrowth of *PTEN* mutant cells under starvation. The eyes get smaller and the growth of the surrounding and the peripheral tissue is restored. The expression of *Dilp-2* has no effect on control eyes. *Figure 7. Continued on next page*

*Figure 7. Continued*

Quantification of eye size (**A′**) and shoulder area (**A″**) from (**A**). (**B**) Systemic reduction of growth by ubiquitous expression of *Imp-L2* with *arm-Gal4* under standard conditions decreases the size of the control eyes but enhances *PTEN* mutant overgrowth. (**B′**) Quantification of eye size and shoulder area from (**B**).

The following source data and figure supplements are available for figure 7:

**Source data 1**. Eye size, shoulder size; ommatidia number; ommatidia size.

**Figure supplement 1**. Growth properties of *PTEN* mutant tissue are influenced by the neighboring tissue.

**Figure supplement 2**. Autonomous and non-autonomous effects of reducing TORC1 activity on the overgrowth of *PTEN* clones.

**Figure supplement 3**. *PTEN* mutant cells neighboring with *TOR* mutant twin clones acquire a proliferative advantage early in larval development.

**Figure supplement 4**. Wild-type cells neighboring with *TOR* or *PKB* mutant cells gain a growth advantage.

**Figure supplement 5**. *PTEN* mutant cells sense the growth reduction in their neighborhood and proliferate faster under starvation.

**Figure supplement 6**. Systemic reduction of growth signaling induces a starvation-like response in *PTEN* mutant cells.

We next tested how the combination of genetically starved neighboring cells (mutant for *TOR*) with real starvation during development impacts on the growth behavior of *PTEN* clones. Interestingly, when in competition with *TOR* mutant cells, *PTEN* clones did overgrow under NR as opposed to normal conditions (*Figure 7—figure supplement 2*). The overgrowth was even more pronounced than without *TOR* mutant neighbors, and it was apparent already early during larval development (*Figure 7—figure supplement 3*). SEM analysis revealed that the additional overgrowth was primarily caused by more cells (*Figure 7—figure supplement 5*).

Finally, we wondered whether systemically dampening IIS would allow the overgrowth of *PTEN* clones under normal conditions. We ubiquitously expressed *Imp-L2*, which encodes a secreted antagonist of Dilp-2 (*Honegger et al., 2008*). In this context, *PTEN* clones did overgrow, irrespective of whether they were neighboring with *TOR* mutant cells or wild-type cells (*Figure 7B*). The overgrowth of the *PTEN* clones caused a further reduction in body size, as evidenced by decreased shoulder width (*Figure 7B*). Cells devoid of *PTEN* also reacted to the decrease in IIS activity (by slightly reducing cell size) but the massive increase in cell number caused the total overgrowth (*Figure 7—figure supplement 6*). Thus, systemically decreasing IIS enhances the proliferative potential of *PTEN* mutant cells.

## Discussion

Tumors with high PI3K pathway activity are associated with increased resistance to starvation. Here, we describe how the loss of the tumor suppressor PTEN contributes to early clonal expansion. *PTEN* mutant cells tolerate and survive starvation in a clonal situation in *Drosophila* imaginal discs. This response is completely dependent on high PKB and sustained TORC1 activities within the *PTEN* clones. *PTEN* mutant cells also acquire a growth advantage under starvation conditions at the expense of wild-type cells in the immediate neighborhood and in the entire organism.

It has previously been demonstrated that activation of PI3K (or of TORC1) results in starvation insensitivity in endoreplicative tissues (ERTs) (*Britton et al., 2002*). The role of activated PI3K in starvation resistance of *Drosophila* epithelia is also not unprecedented. Anaplastic Lymphoma Kinase (ALK)-dependent activation of PI3K is necessary for bypassing the amino acid requirement in growing neuroblasts (neural progenitors) under NR conditions (brain sparing) (*Cheng et al., 2011*). However, imaginal tissues are not spared upon starvation, arguing against an important function of ALK in mediating growth of imaginal discs under NR conditions.

Mitotically active imaginal disc cells with high PIP3 levels respond differently to starvation as compared to corresponding cells in the ERTs. *PTEN* mutant imaginal disc cells do react to starvation by reducing their size like control cells do. The cell size reduction is proportional: *PTEN* mutant cells are still enlarged with respect to control cells under starvation. However, the cell size reduction is

compensated for by a massive increase in proliferation, causing a net increase in mutant tissue. In fact, the *PTEN* mutant tissue is absolutely enlarged under starvation conditions.

*PTEN* mutant cells therefore gain a growth advantage and can take over a complete organ when the surrounding tissue is starved. This advantage is neither due to a prolonged growth period (*PTEN* clones are increased in size early after clone induction) nor to a complete insensitivity to starvation. It rather reflects a change in their mode to respond to a limited access to nutrients: *PTEN* mutant cells switch from hypertrophic to hyperplastic growth under starvation. This tolerance towards increased proliferation could favor the selection for secondary mutations that enhance proliferation and thus tumor progression.

To our surprise, *PTEN* clones display increased levels of apoptosis, and selective inhibition of apoptosis within the clones results in tremendous hyperplastic overgrowth. The induction of apoptosis in *PTEN* mutant tissue contrasts the known pro-survival function of PKB (*Scanga et al., 2000*). However, it indicates that the starved *PTEN* mutant cells, despite their proliferative advantage, exist on the edge of survival and are highly susceptible to apoptosis. Our findings also suggest that acquisition of factors preventing apoptosis would strongly enhance the growth of tumors lacking PTEN.

How can *PTEN* mutant cells grow at the expense of the non-mutant tissue? Comparing the P-PKB levels in the heterozygous and wild-type tissue revealed a reduction in the tissue surrounding *PTEN* clones as compared to tissue adjacent to control clones. Furthermore, cells neighboring with *PTEN* mutant tissue are smaller than those in the discs containing control clones. Thus, *PTEN* clones impact on IIS activity in the surrounding tissue, thereby reducing its growth potential. This appears to be sufficient to explain the strong reduction of the surrounding tissue, since the process of ousting the neighboring tissue is completed as early as 60 hr after clonal induction, when the disc still contains relatively few cells. This non-autonomous growth reduction is not restricted to the local neighbors in a cell–cell contact-dependent manner, but rather affects all tissues of the organism.

The impact of *PTEN* mutant tissue on its surroundings suggests that the cells compete for common factors, most likely growth factors and nutrients. Under starvation, the circulating levels of growth factors and nutrients are strongly reduced, and thereby can become limiting (*Cheng et al., 2011*). *Drosophila* insulin-like peptides (DILPs) produced by the insulin-producing cells (IPCs) in the brain stimulate the growth of imaginal disc cells via the InR. Starvation reduces the secretion of DILPs from the IPCs, and thus reduces growth (*Geminard et al., 2009*). Removing *InR* function in *PTEN* clones only mildly affects the overgrowth. Thus, despite lacking the signal input via InR, these cells are able to accumulate PIP3, probably because of the basal activity of PI3K. Therefore, low levels of circulating DILPs are sufficient to boost PIP3 levels and PKB activity in *PTEN* mutant cells under starvation.

The cationic amino acid transporter Slimfast (Slif) has been described as an upstream activator of TORC1 (*Colombani et al., 2003*). We show that, in contrast to inhibiting TORC1 activity, reducing Slif does not block the initial overgrowth of *PTEN* clones under starvation. However, it causes the overgrown clones to collapse. We speculate that the high metabolic demands of *PTEN* mutant cells require an efficient amino acid uptake, which is blocked by reducing Slif. Thus, *PTEN* mutant cells rely on their ability to efficiently compete for nutrients and growth factors under starvation conditions with their direct neighbors and the peripheral tissue. Since the reduction of Slif affects the growth of the wild-type and the *PTEN* mutant tissue in a differential manner, this amino acid transporter could represent a target for a dosage-dependent drug therapy of tumors with PI3K activation.

Our results suggest the following sequence of events when a single cell embedded in a mitotic tissue loses the tumor suppressor PTEN (*Figure 8*). The loss of *PTEN* function triggers high PIP3 levels, thereby activating PKB and TORC1, thus stimulating cell growth and division. Under starvation conditions, circulating growth factors and nutrients are scarce. In response to these limited resources, both wild-type and *PTEN* mutant cells adjust their growth with respect to cellular size. In sharp contrast to wild-type clones, *PTEN* clones retain high PI3K activity and strongly increase their cell number. This hyperplastic clonal overgrowth depends on withdrawing nutrients from the neighborhood and ultimately the entire organism. The active scavenging of resources is dependent on efficient amino acid transporters (Slif), which also ensure that TORC1 remains active. During their entire growth, cells are at the brink of death as evidenced by the high apoptosis rate within the *PTEN* clones. Thus, enhanced PIP3 levels in cells that lost the tumor suppressor PTEN not only leads to starvation resistance at the cellular level, but also suppresses growth in their direct tissue neighborhood and in peripheral tissues by competing for nutrients and growth factors. Our findings demonstrate how limiting nutrient conditions enhance the proliferative potential of *PTEN* mutant cells, independent of additional genetic alterations.

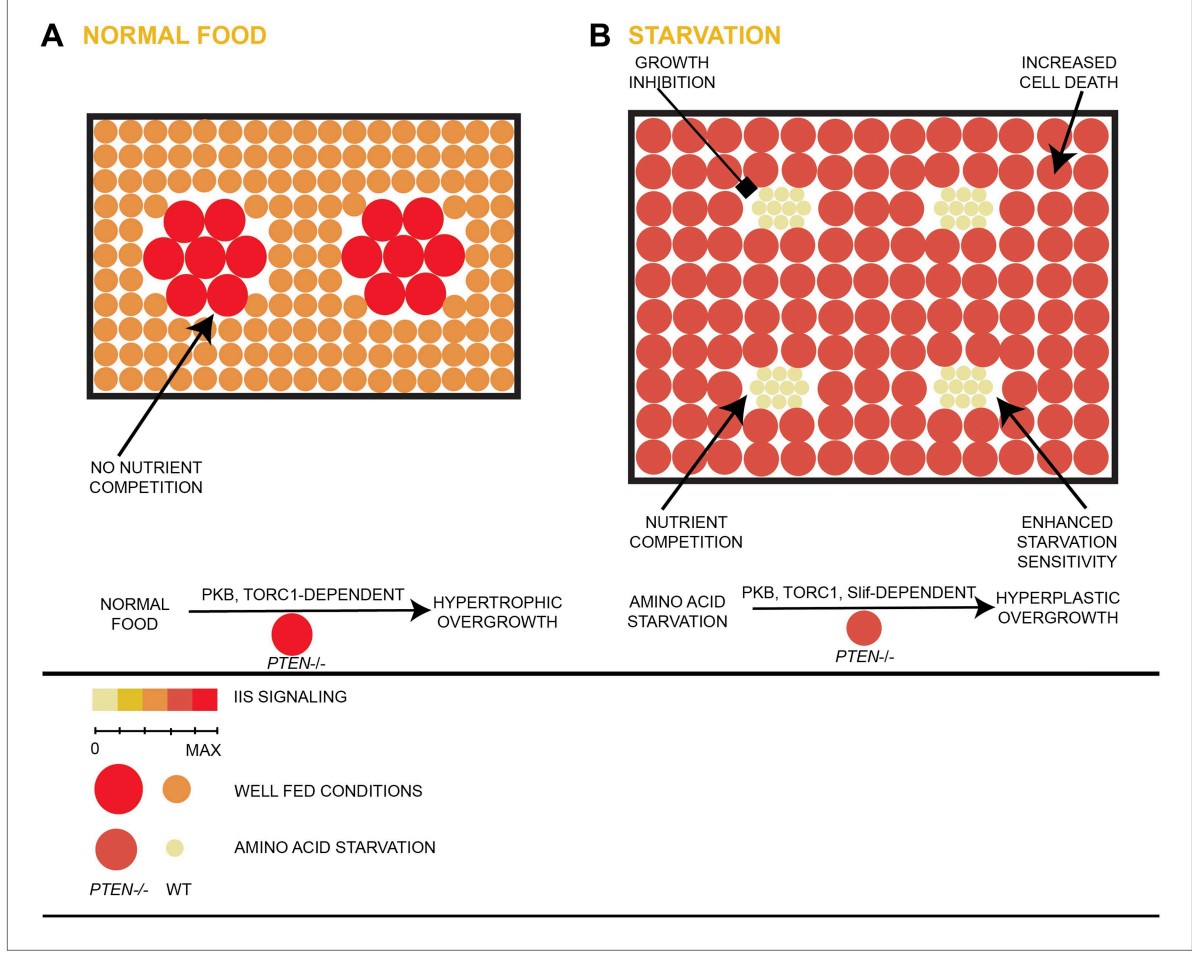

**Figure 8**. Model of hyperplastic overgrowth of *PTEN* mutant tissue under starvation. (**A**) Under standard conditions (normal food), the *PTEN* mutant tissue overgrows in a hypertrophic manner. The tissue is enlarged because of larger *PTEN* mutant cells. (**B**) Under starvation conditions, the *PTEN* mutant tissue is metabolically more active and outcompetes the surrounding wild-type tissue, resulting in a hyperplastic overgrowth. The size of the tissue is further increased because of more *PTEN* mutant cells. The *PTEN* mutant tissue is susceptible to apoptosis, and it depends on the function of the amino acid transporter Slimfast. Under both conditions, the *PTEN* mutant tissue exhibits high insulin signaling activity and is dependent on the functions of PKB and TORC1.

## Materials and methods

### Fly media and stock keeping

1 liter *Drosophila* medium contains 100 g of fresh yeast, 55 g cornmeal, 10 g wheat flour, 75 g sugar and 8 g bacto agar, here referred to as standard medium (or 100 g/l yeast food). Starvation food (or 10 g/l yeast food) was generated by reducing the amount of yeast without altering the other ingredients. All crosses and experiments were performed at 25°C under non-crowding conditions.

### Mutants and transgenes

To generate a viable hypomorphic situation for *PTEN*, the null allele *PTEN117* and the hypomorphic allele *PTEN100A* were used in heteroallelic combinations (*Oldham et al., 2002*). For clonal analysis, *FRT40 PTEN117* and *FRT82 Tsc1Q87X* (*Oldham et al., 2000*; *Tapon et al., 2001*) were used. To generate null mutant clones for *PTEN* on the third chromosome, the genomic rescue construct *FRT82 PTENgenomic rescue ubiGFP* (*Gao et al., 2000*) was used in a null mutant *PTEN117* background. Knockdown of *PTEN* was achieved using the VDRC line 101475 in combination with *GMR-Gal4* and *DE-Gal4* (*Morrison and Halder, 2010*). For genetic interaction studies with the insulin and TOR pathways, the following alleles were used: *FRT82 PKB1* and *FRT82 PKB3* (*Stocker et al., 2002*), *FRT82 InR5545* (*Fernandez et al., 1995*),

FRT82 PI3K92E$^{2H1}$ (**Halfar et al., 2001**), FRT82 FoxO$^{25}$ (**Junger et al., 2003**), FRT FoxO$^{Δ94}$ (**Slack et al., 2011**), FRT40 TOR$^{2L19}$ (**Oldham et al., 2000**), FRT40 TOR$^{EP2353}$ (**Zhang et al., 2000**) and FRT82 Rheb$^{2G5}$ (**Stocker et al., 2003**). The following transgenic fly lines were used: GMR-Gal4 Thor$^{1}$; EP-dFoxO (**Junger et al., 2003**), UAS-cherry dFoxO, UAS-Tsc1/2 (**Tapon et al., 2001**), UAS-Dilp2 (**Brogiolo et al., 2001**), UAS-Imp-L2 (**Honegger et al., 2008**) and UAS-Atg5$^{RNAi}$ (**Scott et al., 2004**). For blocking apoptosis, FRT40 bsk$^{1}$ (**Riesgo-Escovar et al., 1996**), UAS-p35 (**Hay et al., 1995**) and UAS-bsk$^{DN}$ (**Adachi-Yamada et al., 1999**) were used. To monitor PI3K activity, a tGPH reporter was used (**Britton et al., 2002**). The amino acid transporter Slimfast was silenced using Slif$^{anti}$ (**Colombani et al., 2003**). A full list of alleles and transgenes used is provided in **Supplementary file 1**.

## Stress experiments

Flies were crossed in standard rearing vials for 3 days, transferred to laying cages, and allowed to lay eggs on apple agar plates (11 hr in most of the experiments, 3 hr for fat body analysis). For induction of eyFlp/FRT clones, eggs were distributed to the different food conditions immediately, whereas for induction of hsFlp/FRT clones, eggs were allowed to hatch, and they were heat-shocked before distribution. For quantification of survival, dead embryos were counted 24 hr after seeding to the food, and survival of the pupae was recorded.

## Phenotypic analyses

All phenotypic analyses on adult flies were performed on females (unless indicated otherwise), and measurement of nuclear size in fat bodies was done on female larvae. Size and number of the ommatidia in SEM pictures, size of the eyes, shoulders, wings, and fat body nuclei were measured using Photoshop CS3. Student's t-test (two-tailed) was used to test for significance in all the quantification experiments.

For determination of dry weight, flies were dried at 95°C for 5 min and individually weighed with a Mettler Toledo MX5 microbalance.

## Clonal analyses

Mutant clones in eye and wing imaginal discs were generated with y, w, hsFlp; FRT40 or FRT82 flies. Clones were induced during the first instar (heat shock for 15 min at 37°C, 38 hr after egg deposition [AED]), and larvae were dissected in the third instar before wandering, unless otherwise indicated. For positively marked PTEN knockdown clones, Actin Flp-out Gal4 (**Neufeld et al., 1998**) and the MARCM (**Lee and Luo, 2001**) system were used. Clones generated by the Actin Flp-out Gal4 technique were induced during the first larval instar (heat shock for 10 min at 37°C, 38 hr AED), and larvae were dissected in the third instar before wandering. Eye-specific clones were generated using the eyFlp/FRT system. The exact genotypes are indicated in **Supplementary file 1**.

## Immunohistochemistry, Western blotting and histology

Larval imaginal discs were fixed in 4% PFA (30 min, RT), permeabilized in 0.3% PBT (15 min, RT), blocked in 2% NDS in 0.3% PBT (1 hr, RT), incubated with the primary antibodies overnight (4°C), washed three times in 0.3% PBT, and incubated with secondary antibodies (1 hr, RT). The nuclei were visualized with 1:2000 DAPI in 0.3% PBT (15 min, RT). Antibodies used in this study were: rabbit α-Drosophila phospho-Akt/PKB Ser505 (1:300; Cell Signaling), rabbit α-cleaved Caspase 3 (1:300; Cell Signaling), mouse α-β-Galactosidase (1:300; Promega), Cy3- and Cy5-coupled α-mouse or α-rabbit IgG (1:300; Amersham). All the dilutions were made in blocking solution. TUNEL staining was carried out according to the manufacturer's protocol (ApopTag Red In Situ Apoptosis Detection KitS7165; Millipore).

Larval fat bodies were fixed in 8% PFA (45 min, RT) and stained with 1:50 Alexa Fluor 488 phalloidin in 0.2% PBT (90 min, RT; Molecular Probes) and 1:500 DAPI in 0.2% PBT (5 min, RT).

Western blots on L3 eye imaginal discs were performed according to standard protocols. Antibodies were α-Drosophila phospho-PKB Ser 505 (1:1000; Cell Signaling), α-PKB (1:1000; Cell Signaling), α-phospho-S6K (1:1000; Cell Signaling), α-S6K (1:2000; our own antibody), α-Tubulin (1:10,000; Sigma), HRP-conjugated α-mouse and α-rabbit IgG (1:10,000; Amersham).

Histological sections of the adult fly eyes were performed as previously described (**Basler and Hafen, 1988**).

## Image acquisition

For the confocal images, a Leica SPE confocal laser scanning microscope was used. A Jeol JSM-6360LV microscope was used for scanning electron microscope pictures. For color pictures of larvae and

adults, a KEYENCE VHX1000 digital microscope was used. For the pictures of the wings and the histological sections of the eyes, a Zeiss Axiophot Microscope was used.

## Statistical analyses

In all the quantifications, Student's *t*-test (two-tailed) was used to test for significance. In each experiment, a minimum of nine individuals was measured for each genotype. Significance is indicated in the Figures using the following symbols: *p<0.05, **p<0.01, ***p<0.001, n.s., not significant. Error bars represent the standard deviation. All measurement data are provided in the Source Data Files accompanying the Figures including statistical analyses (*Figure 1—source data 1*, *Figure 2—source data 1*, *Figure 3— source data 1*, *Figure 4—source data 1*, *Figure 6—source data 1*, *Figure 7—source data 1*).

## Acknowledgements

We thank P Bansal, M Adlesic, A Baer, J Lüdke, A Straessle, I Vuillez for technical support; K Basler, B Edgar, D Pan, P Leopold, N Tapon, the Bloomington and VDRC stock centers for fly stocks; K Köhler for critical comments on the manuscript; M Aguet, W Krek, P Leopold, J Szabad and present and former members of the Hafen laboratory for discussions.

## Additional information

### Funding

| Funder | Grant reference number | Author |
| --- | --- | --- |
| Swiss National Science Foundation | 31003A_125208 | Hugo Stocker |
| ETH Zürich | | Ernst Hafen |

The funders had no role in study design, data collection and interpretation, or the decision to submit the work for publication.

### Author contributions

KN, GS, Conception and design, Acquisition of data, Analysis and interpretation of data, Drafting or revising the article; EH, Conception and design, Drafting or revising the article; HS, Conception and design, Analysis and interpretation of data, Drafting or revising the article

## Additional files

### Supplementary files

• Supplementary file 1. Genotypes of experimental animals.

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
