## [Decision Letter]

Thank you for choosing to send your work entitled “Nutrient restriction enhances the oncogenic potential of cells lacking the tumor suppressor PTEN in a mitotic tissue” for consideration at *eLife*. Your article has been evaluated by a Senior Editor and 3 reviewers, one of whom is a member of our Board of Reviewing Editors.

The Reviewing editor and the other reviewers discussed their comments before we reached this decision, and the Reviewing editor has assembled the following comments to help you prepare a revised submission.

Substantive concerns to be addressed during revision:

1) The finding that *PTEN* mutant tissues decrease the growth of wild-type tissues is very intriguing, but this aspect of the work is underdeveloped and needs to be investigated at a mechanistic level. First, levels of DILPs and specific nutrients in hemolymph from larvae or adults with *PTEN* loss in the eye should be measured, if possible. Alternatively, for the DILPs, the levels in the median neurosecretory (mNSCs) brain cells that secrete them can be quantitated as a reduction in circulating DILPs – this usually correlates with an increase in DILPs in the mNSCs.

2) Second, it is important to test if a boost in the levels of secreted DILPs is sufficient to rescue the growth defect of wild-type cells in animals with *PTEN*-null clones. In a similar vein, does an increase in Slimfast in wild-type cells ameliorate their growth defects in the context of *PTEN* mutant clones?

3) To expand the scope of the work, it is important to determine if *Tsc1* mutant clones behave similarly to those with reduced PTEN function. Given the results in Figure 6—figure supplement 1, it appears that the necessary strains are available. The results of this experiment would increase the impact of this story by determining whether sustained TORC1 signaling is both necessary and sufficient for the enhanced clonal overgrowth (combined hypertrophy and hyperplasia) under nutrient restriction.

---

## [Author Response]

*1) The finding that* PTEN *mutant tissues decrease the growth of wild-type tissues is very intriguing, but this aspect of the work is underdeveloped and needs to be investigated at a mechanistic level. First, levels of DILPs and specific nutrients in hemolymph from larvae or adults with* PTEN *loss in the eye should be measured, if possible. Alternatively, for the DILPs, the levels in the median neurosecretory (mNSCs) brain cells that secrete them can be quantitated as a reduction in circulating DILPs – this usually correlates with an increase in DILPs in the mNSCs*.

The reviewers raise a key question regarding the unknown mechanism of the observed competition between *PTEN* mutant and normal cells. Indeed, we considered measuring DILP levels. However, as DILP-2, the most potent growth- promoting DILP, is retained in the mNSCs upon starvation, it would be difficult to determine increased retention by immunostaining. For a reliable mass-spectrometric quantification, the amount of tissue obtained from our experimental setup is limiting.

As our genetic experiments pointed to an important role of amino acids, we measured amino acid concentrations in the hemolymph of larvae bearing eye- specific *PTEN* clones and of control larvae by gas chromatography coupled with mass spectrometry (GC-MS). However, the results we obtained are rather complex. It has been reported that the concentrations of alanine, isoleucine, leucine, and valine decrease upon acute starvation (Anaplastic lymphoma kinase spares organ during nutrient restriction in *Drosophila,*
[9]). In our experiment, only two of them, isoleucine and valine, exhibited the same tendency, whereas the concentration of alanine, the most abundant amino acid, even increased upon starvation. Leucine levels were not considerably affected. The results of a representative experiment are shown below in Author response image 1. The discrepancies may be a consequence of the differences in the experimental setups. In our experiment, we subjected the larvae to chronic starvation instead of acute starvation. We checked the changes at the end of a long-term process, which may no longer reflect the situation at the beginning of the starvation. It is also conceivable that changes in amino acid concentrations occur rather locally and not in the systemic pool in the hemolymph.Author response image 1.Concentrations of amino acids in the hemolymph of larvae bearing *PTEN* clones and control larvae under normal and starvation conditions.

We also assessed the starvation response in the fat body, a central organ in nutrient sensing. We performed a gene expression analyses for *InR* and *4EBP-1,* two genes known to be transcriptionally upregulated upon starvation. This experiment is expected to be more sensitive with respect to systemic changes than the measurement of DILPs in the mNSCs by immunostaining. We dissected fat bodies of the tightly staged larvae (egg laying for 5 hours) with *PTEN* mutant and control eye discs early in the third instar (72 h under standard conditions and 120 h under starvation conditions) to avoid any changes in gene expression associated with the metamorphosis processes taking place in the fat body. mRNA levels in fat bodies were determined by qRT-PCR. The levels of *InR* and *4EBP-1* mRNAs were consistently upregulated upon starvation. However, we did not detect any differences between the larvae bearing *PTEN* clones and the control larvae (Author response image 2). Again, changes in gene expression may have been alleviated by adaptation mechanisms occurring during chronic starvation. It is also possible that key events triggering the overgrowth of *PTEN* mutant tissue take place at the onset of starvation.Author response image 2.Expression levels of the starvation-induced genes *InR* and *4EBP-1* in the fat body of control larvae and larvae bearing *PTEN* clones (see text for explanations).

As the quantifications of amino acid levels and of starvation-responsive genes did not yield conclusive results, we resorted to a genetic experiment to support our notion that the growth properties of *PTEN* mutant tissue influence the growth of the surrounding and peripheral tissues, and vice versa. We attempted to genetically starve the larvae in order to mimic the starvation response in the animals. To this end, we systemically dampened IIS by ubiquitously expressing Imp-L2, a secreted antagonist of Dilp2 (by means of *arm-Gal4).* In this context, *PTEN* mutant clones did overgrow more, which was accompanied by a further reduction in body size. Thus, *PTEN* mutant cells compete with the surrounding normal cells for pools of systemic growth factors, and a systemic reduction in IIS enhances the overgrowth of *PTEN* mutant tissue. Closer analysis of the overgrown eyes by SEM revealed that cells devoid of PTEN reacted to systemically decreased IIS by slightly reducing their size, but a massive increase in cell number caused the net overgrowth. Thus, systemically decreasing IIS enhances the proliferative potential of *PTEN* mutant cells. These data are shown in the new Figure 7 and the accompanying Figure 7—figure supplement 6.

*2) Second, it is important to test if a boost in the levels of secreted DILPs is sufficient to rescue the growth defect of wild-type cells in animals with* PTEN*-null clones. In a similar vein, does an increase in Slimfast in wild-type cells ameliorate their growth defects in the context of* PTEN *mutant clones*?

We performed both experiments suggested by the reviewers. Expression of *Dilp2* by means of *ey-Gal4* in the context of *PTEN* clones restored the growth of the surrounding and peripheral tissue and suppressed the *PTEN* overgrowth, while not affecting control eyes. These results are shown in Figure 7. Similar results were obtained by the expression of wild-type *InR* (data not shown).

An analogous experiment was performed with *Slimfast.* Expression of *Slif* (by means of *ey-Gal4)* slightly increased the sizes of control and *PTEN* mutant eyes, but this effect did not reach statistical significance (Author response image 3). Thus, the increased expression of Slif did not restore the growth of the surrounding tissue. However, as we do not know whether increasing the levels of the amino acid transporter Slif enhances its total activity, the experiment is not conclusive and we did not include it in the manuscript.Author response image 3.Overexpression of *Slif* does not rescue the growth of the tissue neighboring with *PTEN* clones.

*3) To expand the scope of the work, it is important to determine if* Tsc1 *mutant clones behave similarly to those with reduced PTEN function. Given the results in*
Figure 6—figure supplement 1*, it appears that the necessary strains are available. The results of this experiment would increase the impact of this story by determining whether sustained TORC1 signaling is both necessary and sufficient for the enhanced clonal overgrowth (combined hypertrophy and hyperplasia) under nutrient restriction*.

We agree that the analysis of *Tsc1* mutant clones is an important issue. Cells devoid of Tsc1 are overgrowing under starvation conditions, similar to *PTEN* mutant cells. However, a thorough analysis revealed significant differences between *PTEN* and *Tsc1* clones. As this story is rather complex, we did not want to include it in the present manuscript.

Acknowledgement: The authors thank Manuel Hörl and Nicola Zamboni (Institute of Molecular Systems Biology, ETH Zürich) for the GC-MS analyses shown in Author response image 1.